

# Impact of artificial waterholes on temporal partitioning in a carnivore guild: a comparison of activity patterns at artificial waterholes to roads and trails

Charlotte Krag[1], Linnea Worsøe Havmøller[1], Lourens Swanepoel[2,3], Gigi Van Zyl[4], Peter Rask Møller[1] and Rasmus Worsøe Havmøller[1,5]

[1] Department of Zoology, Natural History Museum of Denmark, University of Copenhagen, Copenhagen, Denmark
[2] Department of Biological Sciences, Faculty of Science, School of Engineering and Agriculture, University of Venda for Science and Technology, Thohoyandou, Limpopo Province, South Africa
[3] DNRF-SARChI Chair in Biodiversity Value and Change, University of Venda for Science and Technology, Thohoyandou, South Africa
[4] Maremani Game Reserve, Musina, Limpopo Province, South Africa
[5] Department of Migration, Max Planck Institute for Animal Behaviour, Radolfzell am Bodensee, Baden-Württemberg, Germany

Corresponding author
Rasmus Worsøe Havmøller,
RGHavmoller@snm.ku.dk

## ABSTRACT

Temporal partitioning in large carnivores have previously been found to be one of the main factors enabling co-existence. While activity patterns have been investigated separately at artificial waterholes and *e.g.*, game trails, simultaneous comparative analyses of activity patterns at artificial waterholes and game trails have not been attempted. In this study, camera trap data from Maremani Nature Reserve was used to investigate whether temporal partitioning existed in a carnivore guild of four species (spotted hyena, leopard, brown hyena and African wild dog). Specifically, we investigated temporal partitioning at artificial waterholes and on roads and trails an average of 1,412 m away from an artificial waterhole. Activity patterns for the same species at artificial waterholes and roads/game trails were also compared. We found no significant differences in temporal activity between species at artificial waterholes. Temporal partitioning on game trails and roads was only found between spotted hyena (nocturnal) and African wild dog (crepuscular). Between nocturnal species (spotted hyena and leopard) no temporal partitioning was exhibited. Only African wild dog exhibited significantly different activity patterns at waterholes and roads/game trails. This indicates artificial waterholes may be a location for conflict in a carnivore guild. Our study highlights the impact of anthropogenic landscape changes and management decisions on the temporal axis of carnivores. More data on activity patterns at natural water sources such as ephemeral pans are needed to properly assess the effect of artificial waterholes on temporal partitioning in a carnivore guild.

## INTRODUCTION

Carnivores are in decline worldwide, with African wild dogs (*Lycaon pictus*) having suffered a loss of over 90% of their historical distribution, and leopards (*Panthera pardus*) experiencing a reduction of nearly 80% of their former range (*Jacobson et al., 2016*; *Wolf & Ripple, 2017*). To make informed and science-based conservation and wildlife management decisions, a comprehensive understanding of the interactions between these large carnivores in relation to key resources is imperative. In many arid-to semi-arid savannas water sources deplete as the dry season progresses. To sustain year-round availability of water to wildlife, artificial waterholes have become increasingly common in protected areas in southern Africa (*Biggs & Rogers, 2003*). Artificial waterholes reduce dry-season wildlife mortality, attract animals for game viewing or hunting and can alter large herbivore presence (*Child, 1972*; *Owen-Smith, 1996*; *Chamaillé-Jammes, Valeix & Fritz, 2007*). Despite proven benefits of year-round water availability, water provisioning has several negative impacts, which include; overutilisation of the surrounding vegetation (*Smit, Grant & Devereux, 2007*; *Mukaru, 2009*), change in spatial patterns of animals (*Valeix, Chamaillé-Jammes & Fritz, 2007*; *Purdon & Van Aarde, 2017*), poison vulnerability from poachers or retributions (*Mzumara, Perrin & Downs, 2016*; *Ogada, Botha & Shaw, 2016*) and increase in interspecific conflicts and competition between species (*Thrash, Theron & Bothma, 1995*; *Valeix, Chamaillé-Jammes & Fritz, 2007*). For example, in Hwange National Park in Zimbabwe water provisioning has caused prey species to change behaviour around water sources when the risk of encountering lions (*Panthera leo*) is high (*Valeix et al., 2009*). To reduce the effect of interspecific conflict, a subordinate competitor may avoid contact with a more dominant competitor by partitioning of the shared niche, such as dietary, spatial, and temporal partitioning (*Schoener, 1974*; *Carothers & Jaksić, 1984*; *Durant, 2000*).

Resource partitioning is an effective strategy for avoiding negative contact with a competitor when the resource is not limited or fixed and is used by many species (*Schoener, 1974*; *Carothers & Jaksić, 1984*; *Durant, 2000*). In the studies of *Hayward & Slotow (2009)* and *Hayward & Hayward (2007)* it was found that subordinate members of a large carnivore guild, African wild dogs and cheetahs (*Acinonyx jubatus*), used temporal partitioning to minimise overlap with dominant species, like lions and spotted hyenas (*Crocuta crocuta*). However, when the resource is spatially fixed, as in the case of artificial waterholes, it can lead to interference competition (*Vahl, 2006*) and spatial partitioning may be limited. *Sirot, Renaud & Pays (2016)* found that herbivores compete heavily for water during daylight to avoid predation and, under conditions of spatially fixed and limited resources, changing the temporal activity pattern may be the only solution to avoid direct competition. According to their results, the competition between herbivores during daylight becomes so strong that it forces some prey species to drink at night when there is a higher risk of predation (*Sirot, Renaud & Pays, 2016*).

Artificial waterholes have previously been associated with impacts on vegetation and alteration of herbivores spatial movements (*Smit, Grant & Devereux, 2007*; *Mukaru, 2009*; *Purdon & Van Aarde, 2017*), and provide an ideal model system to investigate how

spatially fixed resources affect carnivore temporal activity patterns (*Atwood, Fry & Leland, 2011*; *Edwards, Gange & Wiesel, 2015*). Especially of interest would be if carnivores exhibit temporal partitioning at artificial waterholes. If not, artificial waterholes could facilitate interference competition, which could possibly lead to the displacement of subordinate species. This level of competition might not be problematic for some carnivore species, but it could negatively impact rare or endangered species such as African wild dogs.

African wild dogs display crepuscular behaviour, exhibiting the highest level of activity during early morning and early evening hours (*Hayward & Slotow, 2009*; *Woodroffe, 2011*; *Dröge et al., 2017*). In contrast, leopards exhibit a more nocturnal activity pattern, while both spotted- and brown hyenas (*Parahyaena brunnea*) are predominantly nocturnal; with limited activity during daylight hours (*Hayward & Slotow, 2009*; *Vissia, Wadhwa & van Langevelde, 2021*; *Vissia, Fattebert & van Langevelde, 2022*).

Camera traps have been used in multiple studies to investigate overlap in temporal activity patterns within and between species of large African carnivores (*Havmøller et al., 2020a*; *Searle et al., 2021*; *Vissia, Fattebert & van Langevelde, 2022*). A main point of critique of using camera traps in describing activity patterns, however, is that they only record activity at the exact place where they are deployed (*Rovero & Zimmermann, 2016*). While this limitation can be mitigated with random and exhaustive sampling, it can also be exploited to explore ecological questions. For example, visitation by a target species at sites can be random, but it is generally dependent of various factors. These factors can include resources (*e.g.*, carcass), latrines, baits or lures or travel routes. In this study, we explored site use by carnivores at two distinct point localities–first, waterholes and second, travel routes (roads). This setup allowed us to simultaneously evaluate temporal overlap at roads and waterholes. We used camera trap datasets from Maremani Nature Reserve, an arid to semi-arid tropical savanna in South Africa with numerous artificial waterholes, to investigate the temporal overlap within a guild of four carnivore species: African wild dog, leopard, spotted hyena, and brown hyena. We predict that the limited opportunity for spatial partitioning at a fixed key resource will result in significant segregation in the use of the dial cycle between African wild dogs, leopards and spotted hyenas at artificial waterholes. Conversely, we predict that the presence of available spatial and dietary partitioning opportunities on roads and game trails will result in similar activity patterns among leopards, spotted hyenas and brown hyenas (*Vissia, Fattebert & van Langevelde, 2022*). We predict that African wild dogs, however, will exhibit a significantly different activity pattern compared to the other species in the guild, given their documented tendency towards temporal partitioning in the presence of more dominant carnivores and their physiological adaptations towards diurnal behaviour (*Hayward & Slotow, 2009*; *Vanak et al., 2013*; *Dröge et al., 2017*).

## MATERIALS AND METHODS

### Study area

The Maremani Nature Reserve (MNR) is a 40,000-ha fenced arid to semi-arid tropical savanna located in northernmost South Africa close to the Limpopo River, and bordering Zimbabwe (22°23′30.4″S 30°12′55.2″). The reserve was developed by the Aage V. Jensen

Charity Foundation in 1999 to ensure the conservation and protection of wildlife in the area through the acquisition of a number of arid hunting and infertile farming areas (*Maremani Nature Reserve, 2004*).

The mammal community has been re-established through reintroduction of extirpated species, except for lion, cheetah, black rhinoceros (*Diceros bicornis*) and white rhinoceros (*Ceratotherium simum*). According to historical records, brown hyenas and African wild dogs were considered to be abundant, while no historical documentation exists regarding the population sizes of spotted hyenas and leopards (*Maremani Nature Reserve, 2004*). Spotted hyenas began to appear in Maremani in late 2015. Despite a historical absence, recent observations and aerial surveys have indicated that both spotted hyenas and leopards are now increasing in population size on the reserve. On the other hand, brown hyenas appear to have experienced a decline in numbers since 2016 (R. Botha, 2023, personal communication). African wild dogs have been resident in Maremani Nature Reserve since 2013, with their first successful breeding in 2019 and every year since (R. Botha, 2023, personal communication). A single free-roaming pack of wild dogs currently roams Maremani, although another pack resides on the other side of the Limpopo River in Zimbabwe (R. Botha, 2023, personal communication).

The area is characterised by plains, undulating hills, rocky outcrops, higher mountain ranges, rivers, and streams. The altitude varies from 427 m along the Limpopo River to 833 m at Mount Ha-Dowe. The mean annual rainfall for the Maremani area varies from 331 mm at Messina in the west to 342 mm at Tshipise in the south (*Van Rooyen, 2002*). The rainy season is predominantly from October to March with about 85% of the mean annual rainfall occurring during these months. The driest months are from June to August. Besides two rivers, the Sand and Nzhelele river, and natural springs, MNR provides its wildlife with artificial water sources to ensure year-round availability. During our study period a total of 27 functional waterholes were available to wildlife in MNR (Fig. 1). The artificial water holes are intended for wildlife only, and water is extracted from boreholes using either generator- or windmill-powered pumps into ground level troughs, differing in size and design (*Sutherland, Ndlovu & Pérez-Rodríguez, 2018*). These artificial waterholes are closed and reopened to manage animal densities and their potential negative effect on the vegetation (*Maremani Nature Reserve, 2004*).

## Data collection and statistical analyses

We gathered camera trap data from three separately independent surveys (Fig. 1). Survey 1 sampled data during the late dry season to late wet season from October 2019 to April 2020 using 34 camera trap stations with Xenon flash Cuddeback Ambush camera traps (Cuddeback Non Typical Inc., Istanti, WI, USA). Survey 2 sampled data during the late dry season from September to November 2021 using 31 camera trap stations with Xenon flash Cuddeback Ambush camera traps (Cuddeback Non Typical Inc., Istanti, WI, USA). Survey 3 sampled data throughout the year from December 2012 to January 2022. In this survey 24 camera trap stations (Covert NBF 30, Covert Code Black; Covert Scouting Cameras Inc., Lewisburg, KY, USA) and Bushnell HD (Bushnell corporation, Overland Park, KS, USA) were placed at artificial waterholes.
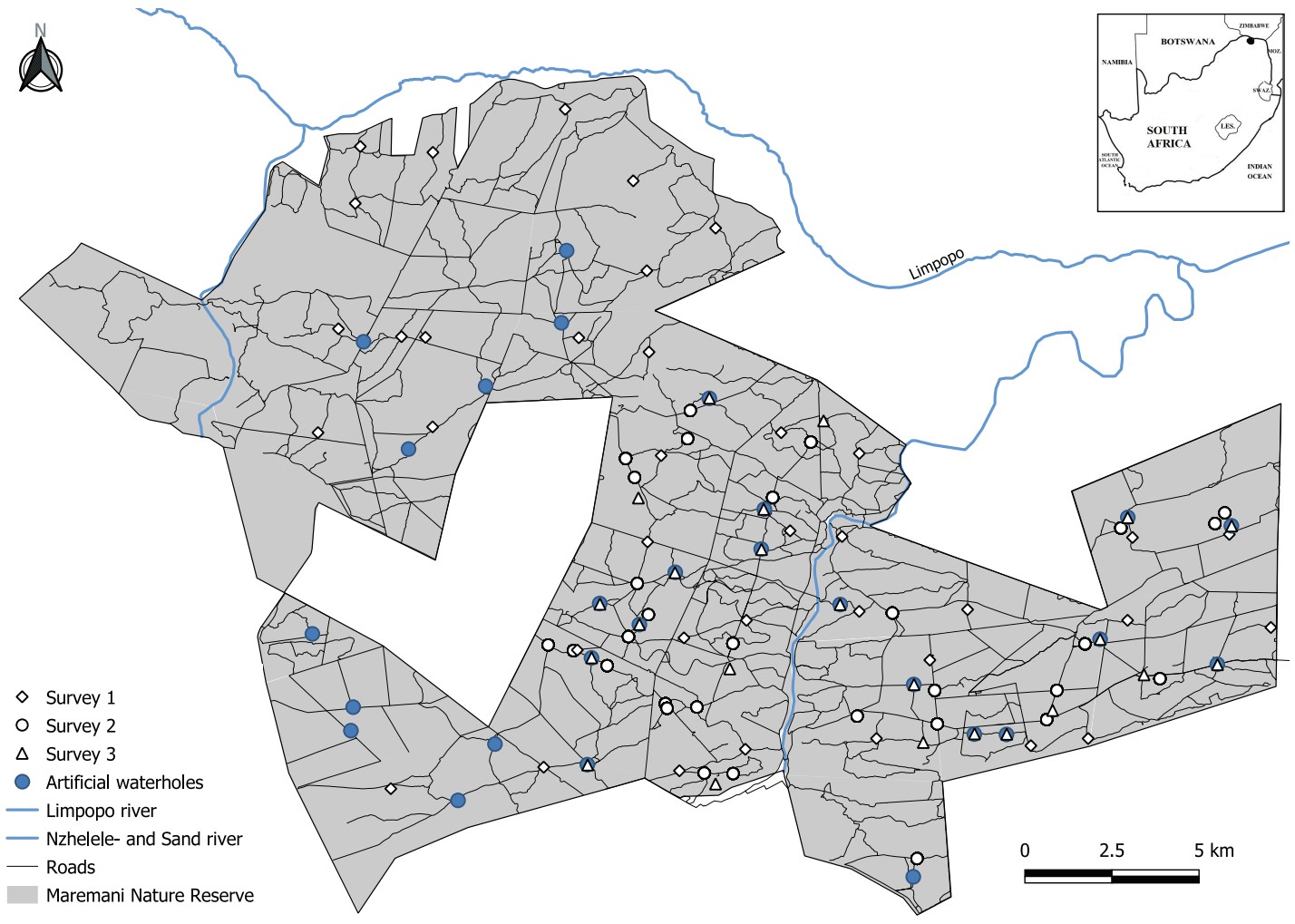

**Figure 1** **Map of Maremani Nature Reserve located in the northernmost part of South Africa bordering the Limpopo River and Zimbabwe.** Camera trap locations on roads and trails (survey 1 and 2), and by artificial waterholes (survey 3). Dirt roads illustrated by thin black lines (MNRroads).

We divided the camera trap data into two categories: (1) Data captured at artificial waterholes and (2) data captured an average of 1,412 m (min 270 m, max 6,042 m) away from an artificial waterhole on roads and game trails (Fig. 1). The minimum acceptable distance for a camera trap record to be considered unassociated with a visit to a waterhole was set to 270 m as this distance constitute ~25% of the lowest minimum daily distance moved by leopards (*Havmøller et al., 2019*) and nearly three times the mean hourly displacement by leopards in *Habib et al. (2021)*. Camera traps placed at an artificial waterhole were facing the water's edge. Camera traps on roads and game trails were non-paired and placed perpendicular on roads and on game trails at a height of 30 cm. Despite the discrepancy in survey duration between the two categories, we argue that the combined data from survey 1 and 2 are comparable to the data obtained from survey 3, as both datasets encompass observations in both the wet and dry seasons. We deemed this a critical factor for ensuring compatibility when we study the effect of artificial waterholes.

Images retrieved from the camera traps were annotated through Wild.ID v.1.0.1 (*Ahumada et al., 2020*) and then analysed in R software version 4.0.3 (*R Core Team, 2020*) using the 'overlap' package v.0.3.4, (*Meredith & Ridout, 2021*) and 'Activity' package v.1.3.2 (*Rowcliffe, 2022*). We used camera trap data for leopards, brown hyenas, spotted hyenas and wild dogs. Unfortunately, records of brown hyenas at artificial waterholes were deleted without being annotated. This was due to them being a very prevalent species at the time of data collection and a lack of recognition of their conservation significance, resulting in the regrettable deletion of the data. Records of brown hyenas are therefore only found in the dataset on roads and game trails. A Hermans–Rasson test was performed for each species to assess if a random activity pattern was exhibited over a circadian cycle to ensure that the paired analyses consisted of clustered datasets (*Landler, Ruxton & Malkemper, 2019*).

We investigated temporal overlap for three different categories. First, we investigated temporal overlap between species on roads and game trails. Second, we investigated temporal overlap between species at artificial waterholes; and third, we investigated temporal overlap for the same species at artificial waterholes and on roads and game trails.

To estimate overlap we used the method by *Ridout & Linkie (2009)*. We set an interval of 30 min between images to adhere to independent assumptions (*Linkie & Ridout, 2011*). Extracted time from photographs were converted to radians and used to fit a non-parametric circular kernel-density function (*Meredith & Ridout, 2021*). An overlapping coefficient ($\Delta$) was then used to measure the extend of overlap between the kernel-density estimate of the different species. The overlap coefficient ranges from 0 (no overlap) to 1 (complete overlap) (*Ridout & Linkie, 2009*; *Linkie & Ridout, 2011*). For larger number of detections (>75 camera records) we used $\Delta 4$ and $\Delta 1$ for smaller samples (*Meredith & Ridout, 2021*). Bootstrap percentile confidence intervals were calculated, based on 10,000 bootstrap samples. The significance of pairwise comparisons between activity levels, were estimated by a Wald test using the 'CompareAct' function provided in the package 'activity' (*Rowcliffe et al., 2014*).

## RESULTS

Survey 1 had a cumulative effort of 5,941 camera trap days while survey 2 had a cumulative effort of 1,432 camera trap days. Combined, survey 1 and 2 resulted in 751 independent events of our four focal species (Table 1). Survey 3 at waterholes from year 2012 until year 2022 resulted in a total of 2,642 independent events for three out of four of our focal species (Table 1).

All species had activity patterns that were significantly different from random according to the Hermans–Rasson test (Table 1). African wild dogs were the only species with a significant difference in activity patterns between artificial waterholes and on roads and game trails (Fig. 2; Table 2). All species exhibited predominantly nocturnal behaviour on roads and game trails, except for African wild dogs which exhibited a crepuscular bimodal activity pattern with peaks around sunrise and sunset (Fig. 3).

On roads and game trails African wild dogs exhibited significantly different activity patterns compared to the rest of the carnivore guild in this study (Table 2). For all other

**Table 1 Summary of number of independent events captured for target species at artificial waterholes and away from artificial waterholes.**

| Common name | Scientific name | N at artificial waterholes | N roads and game trails | Hermans-Rasson test | |
| --- | --- | --- | --- | --- | --- |
| | | | | P-value at artificial waterhole | P-value roads and game trails |
| Leopard | *Panthera pardus* | 1,382 | 232 | 0.001 | 0.001 |
| Spotted hyena | *Crocuta crocuta* | 1,229 | 115 | 0.001 | 0.001 |
| African wild dog | *Lycaon pictus* | 31 | 43 | 0.005 | 0.001 |
| Brown hyena | *Parahyaena brunnea* | * | 361 | – | 0.001 |

Notes:
This includes a summary of Hermans–Rasson uniformity tests made to assess if a random activity pattern was exhibited over a circadian cycle at artificial waterholes and on roads and trails for leopard, spotted hyena, brown hyena and African wild dog derived from camera trapping in Maremani Nature Reserve, South Africa. All species had activity patterns that were significantly different from random according to the Hermans–Rasson test at artificial waterholes and away from artificial waterholes. *N* is the number of independent events per species (minimum 30-min intervals between events).
*Brown hyenas used to be extremely common at artificial waterholes and were not a species of interest, consequently data on them was deleted without being annotated.

species pairings between leopards, spotted hyenas and brown hyenas no significant difference in activity patterns on roads and game trails were observed (Table 2).

At artificial waterholes spotted hyenas and leopards exhibited mainly nocturnal behaviour but with peaks around sunrise and sunset (Fig. 4). African wild dogs exhibited crepuscular behaviour with all their activity centred around sunrise, declining towards midday, and increasing again before sunset (Fig. 4). At artificial waterholes, none of the other species' pairings had significantly different activity patterns from themselves or each other (Table 2).

# DISCUSSION

While previous studies have examined temporal partitioning of carnivores around artificial waterholes (*Atwood, Fry & Leland, 2011*; *Hayward & Hayward, 2012*; *Edwards, Gange & Wiesel, 2015*) and activity patterns at locations unrelated to artificial waterholes (*Hayward & Slotow, 2009*; *Havmøller et al., 2020a*; *Vissia, Wadhwa & van Langevelde, 2021*; *Evers et al., 2022*), our study aimed to combine these approaches and investigate temporal partitioning of a carnivore guild at artificial waterholes and along roads and game trails simultaneously. The targets in our study were the large carnivore guild in Maremani Nature Reserve, consisting of leopard, spotted hyena, brown hyena and African wild dog. Using camera trap data we could compare the activity patterns of the same species at artificial waterholes to activity patterns of those on roads and game trails away from this critical resource.

## Activity patterns at waterholes

We found that African wild dogs were the only species that had a significant difference in activity pattern at artificial waterholes and on roads and game trails. African wild dogs had their main peak in activity on roads and game trails in the morning around 06:00, which coincides with their most frequent hunting times found by *Dröge et al. (2017)*. At artificial waterholes, wild dogs peak later between 07:00–10:00, which could indicate that they drink after hunting in the early mornings. At artificial waterholes wild dogs did not have significantly different activity patterns to leopards and spotted hyenas, however, wild dogs

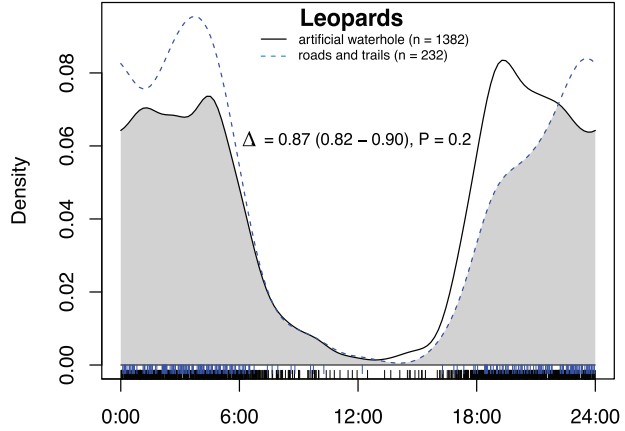

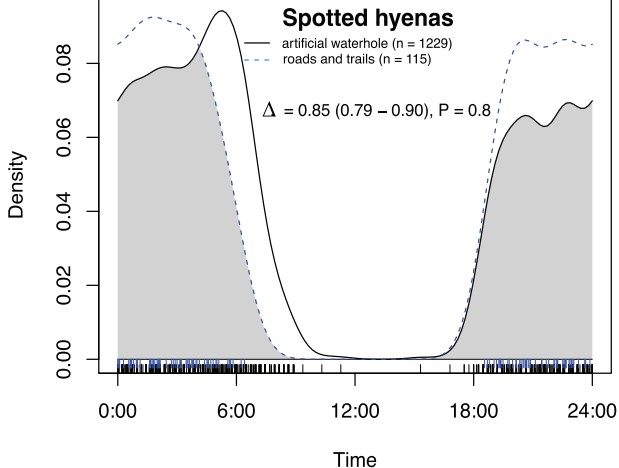

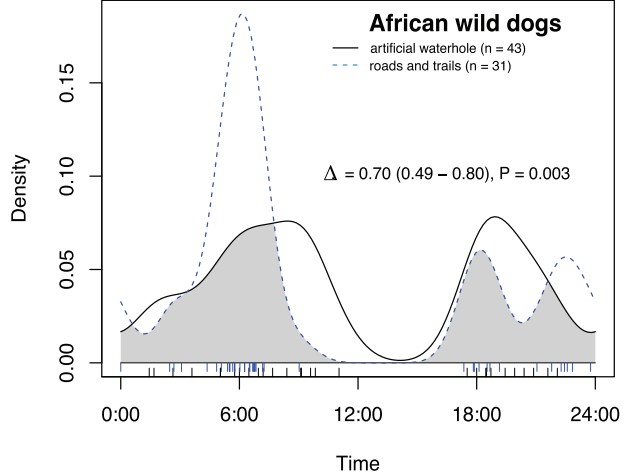

**Figure 2 Overlap in activity pattern within the same species at and away from artificial waterholes.** Data collected from camera trap data collected on Maremani Nature Reserve, South Africa. Coefficient of overlap (Δ), confidence intervals and *P*-values from a test of probability that two sets of circular observations come from the same distribution. African wild dog exhibit significantly different activity

**Figure 2** (continued)
patterns at and away from artificial waterholes. Leopard and spotted hyena do not exhibit significantly different activity patterns at artificial waterholes to their own activity patterns away from artificial waterholes.     

**Table 2  Summary of the Wald test to test for significant differences in a pairwise comparison of activity levels.**

| Overlap on roads and game trails | Δ | lcl | ucl | *P*-value |
|---|---|---|---|---|
| Leopard *vs* Brown hyena | 0.82 | 0.76 | 0.87 | 0.1 |
| Leopard *vs* Spotted hyena | 0.87 | 0.82 | 0.95 | 0.8 |
| Leopard *vs* African wild dog | 0.64 | 0.47 | 0.74 | **<0.001** |
| Spotted hyena *vs* African wild dog | 0.56 | 0.38 | 0.65 | **<0.001** |
| Spotted hyena *vs* Brown hyena | 0.88 | 0.83 | 0.96 | 0.06 |
| African wild dog *vs* Brown hyena | 0.51 | 0.34 | 0.60 | **0.01** |
| **Overlap at artificial waterholes** | | | | |
| Leopard *vs* Spotted hyena | 0.85 | 0.82 | 0.89 | 0.1 |
| Leopard *vs* African wild dog | 0.66 | 0.54 | 0.81 | 0.9 |
| Spotted hyena *vs* African wild dog | 0.63 | 0.48 | 0.76 | 0.5 |
| **Overlap within species at artificial waterholes and on roads and game trails** | | | | |
| Leopard | 0.87 | 0.82 | 0.90 | 0.2 |
| Spotted hyena | 0.85 | 0.79 | 0.90 | 0.8 |
| African wild dog | 0.70 | 0.49 | 0.80 | **<0.01** |

Note:
All species pairings away from artificial waterholes with African wild dog showed a significant difference in activity patterns. All species pairings at artificial waterholes showed no significant difference in activity patterns. Only African wild dogs exhibited different activity patterns at and away from artificial waterholes. Numbers in bold indicate statical significant values.

and leopards had a high overlap in activity at artificial waterholes in the evenings, whereas spotted hyenas rarely appeared before after sunset. These results contradict evidence from *Atwood, Fry & Leland (2011)* and *Edwards, Gange & Wiesel (2015)*, who recorded temporal partitioning between carnivores at artificial water points in dry environments in USA and Namibia. Our findings suggest that, within the context our study and site, temporal resource partitioning is not the main driver for coexistence of a carnivore guild at artificial waterholes. The absence of temporal partitioning at artificial waterholes could increase the risk of interspecific conflict. Therefore, this study suggests that artificial waterholes contained in a fenced arid to semi-arid savanna reserve have the potential to become locations of competition, which could threaten lesser dominant species such as the African wild dog. This could provide a dilemma in areas where permanent water sources are needed in dry seasons to maintain herbivore numbers.

## Activity patterns on roads and game trails
African wild dogs exhibited a significantly different activity pattern on roads and game trails compared to the three other species investigated. On roads and game trails, we found temporal partitioning between African wild dogs and their species pairings with leopards, spotted hyenas, and brown hyenas. This pattern is similar to those observed by *Hayward &*

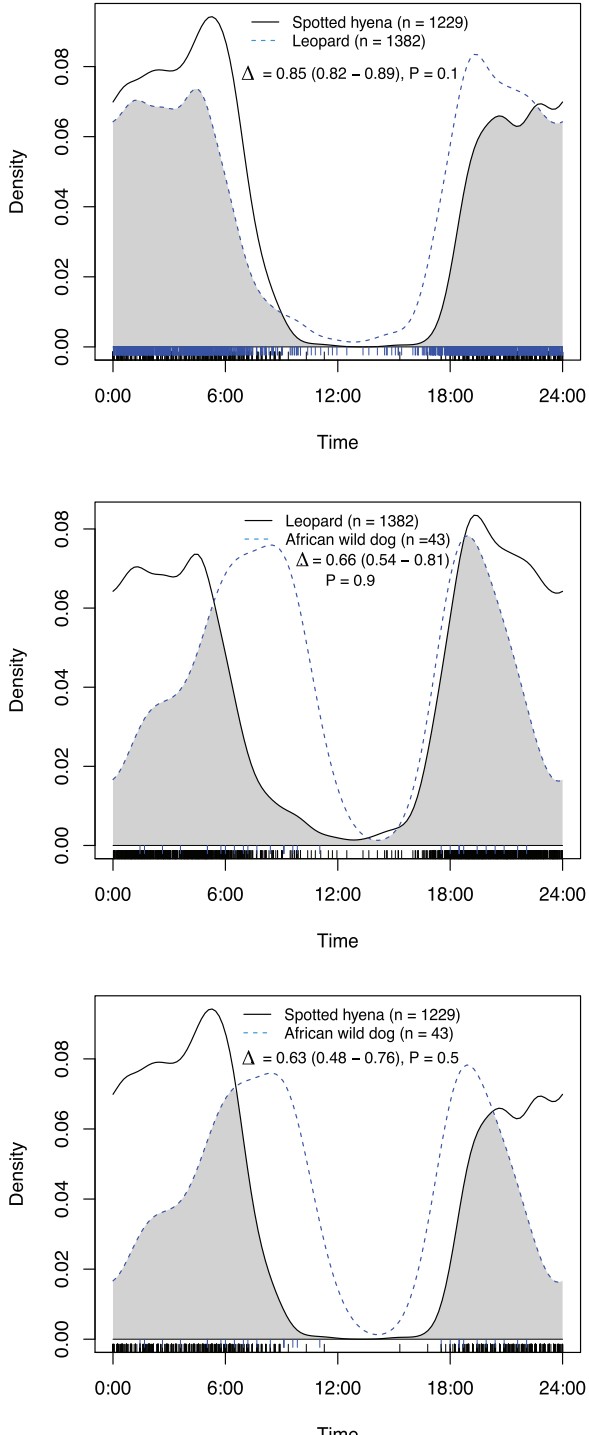

**Figure 3 Overlap in activity pattern of three species: leopard, spotted hyena and African wild dog, at artificial waterholes.** Data collected from camera trap data collected on Maremani Nature Reserve, South Africa. Coefficient of overlap (Δ), confidence intervals and *P*-values from a test of probability that two sets of circular observations come from the same distribution. All species pairings exhibit no significantly different activity patterns from each other at artificial waterholes.

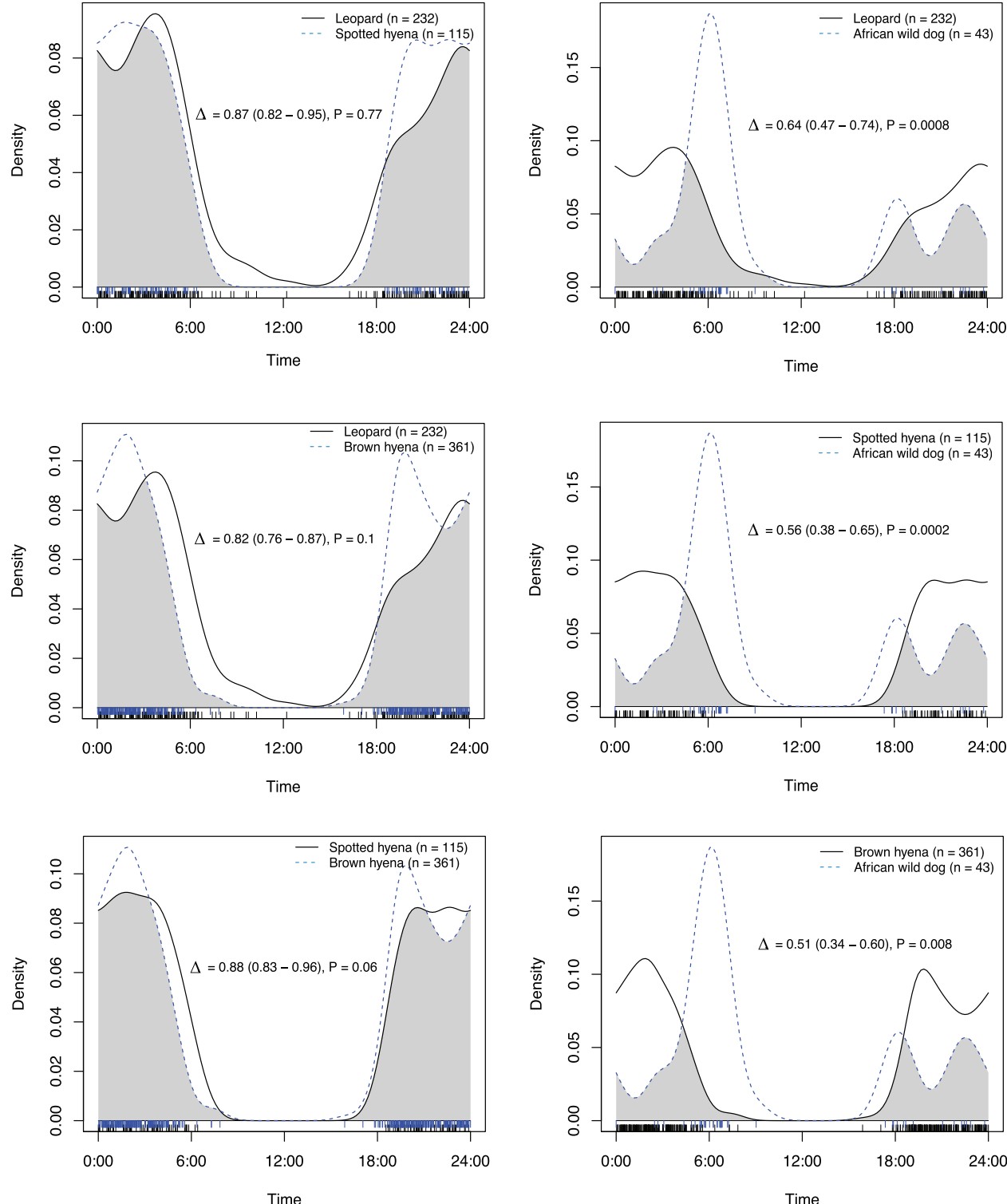

**Figure 4 Overlap in activity pattern in a carnivore guild of four species; leopard, spotted hyena, brown hyena and African wild dog, on roads and trails min.** 270 m away from an artificial waterhole. Data from camera trap data collected on Maremani Nature Reserve, South Africa. Coefficient of overlap ($\Delta$), confidence intervals and *P*-values from a test of probability that two sets of circular observations come from the same distribution. African wild dog exhibits a crepuscular bimodal activity pattern which is a significantly different activity patterns from the rest of the guild. Leopard, spotted hyena, and brown hyena exhibit nocturnal behaviour and do not have significantly different activity pattern from that of each other.

*Slotow (2009)*, who found that subordinate carnivore species, like African wild dog, exhibit temporal partitioning from dominant intraguild predators such as lions and spotted hyenas.

On roads and game trails, spotted hyenas, leopards and brown hyenas all showed no significant difference in activity patterns which is similar to the findings of *Vissia, Fattebert & van Langevelde (2022)*, *Searle et al. (2021)* and *Evers et al. (2022)*. This temporal overlap may be associated with drier open habitat types as *Havmøller et al. (2020a)* have otherwise shown a strict temporal partitioning between leopards and spotted hyenas in the rainforest of the Udzungwa mountains in Tanzania. Other factors influencing temporal partitioning between leopards and spotted hyenas could be the presence of lions as in the study of *Evers et al. (2022)*, or anthropogenic factors such as fencing, which was not present at the study site in *Havmøller et al. (2020a)*.

Brown hyenas and spotted hyenas have previously been found to have a high temporal overlap (*Vissia, Wadhwa & van Langevelde, 2021*), which is also seen in our study. This overlap in activity could possibly be explained by a dietary partitioning where brown hyenas are known to mainly scavenge whereas spotted hyenas are known to also hunt and kill their own prey (*Stein, Fuller & Marker, 2013*; *Yarnell et al., 2013*; *Mills, 2015*).

## General activity patterns

In our study site, African wild dogs exhibit a crepuscular activity pattern compared to the nocturnal behaviour of leopards, spotted hyenas and brown hyenas. In the case of wild dogs, the temporal segregation can be attributed to their need to focus on time windows that minimise night-time hunting by other members of the guild (*Dröge et al., 2017*). However, during nights with ample moonlight, they may exhibit overlap in time with their competitors driven by an increased hunting success rate (*Cozzi et al., 2012*).

The pack of African wild dogs documented at our study site are free roaming, meaning they traverse in and out of fenced and protected areas, and into neighbouring areas where they are unwanted. Consequently, instances of human persecution and killings of pack members have been reported. To avoid persecution from humans, African wild dogs have been shown to shift their activity pattern to become more nocturnal (*Cozzi et al., 2012*; *Rasmussen & Macdonald, 2012*). However, our data suggest that if they were to shift towards a more nocturnal activity pattern it would increase their temporal overlap with other large carnivores.

The lack of temporal segregation among the other members of the carnivore guild may result from a combination of different segregation strategies and physiological constraints. A study conducted in the Cederberg mountains of South Africa revealed that leopards demonstrated increased hunting success during darker nights (*Martins & Harris, 2013*). Their nocturnal behaviour, likely driven by hunting success, makes their activity pattern overlap with both spotted- and brown hyenas (*Vissia, Fattebert & van Langevelde, 2022*). Both spotted- and brown hyenas are suspected to follow leopards in order to kleptoparasite their prey (*Vissia, Fattebert & van Langevelde, 2022*). As an alternative method of segregation leopards have been observed to feed on different sized prey (*Hayward &*

*Kerley, 2008*; *Voigt et al., 2018*; *Havmøller et al., 2020b*), and avoid confrontations and kleptoparasitism from larger predators by caching their prey in trees (*Balme et al., 2017*).

The nocturnal behaviour of spotted hyenas is not solely a function of their need for darkness to hunt successfully, as is the case with leopards, but rather a result of their need to avoid high temperatures (*Cooper, 1990*; *Hayward & Hayward, 2007*). Furthermore, spotted hyenas possess a broad dietary niche, allowing them to target alternative prey and avoid competition with other members of the carnivore guild (*Hayward & Kerley, 2008*).

## Complexity of the carnivore guild: dominant *vs* subordinate members

The African savannah carnivore guild is a complex system with interspecies interactions between multiple large carnivores (*Vanak et al., 2013*). These interspecies interactions may be affected by a range of factors (*e.g.*, habitat, activity of preferred prey, hunting strategy, sensitivity to ambient temperature and composition of the local carnivore guild) (*Creel, 2001*; *Hayward & Hayward, 2007*; *Havmøller et al., 2020b*). African wild dogs are generally considered subordinate to lions (*Hayward & Slotow, 2009*) and according to *Vanak et al. (2013)* they even avoid leopards and cheetahs. However, their relationship with spotted hyenas may depend on the group sizes of both species (*Creel & Creel, 1996*; *Darnell et al., 2014*; *Dröge et al., 2017*).

Maremani Nature Reserve does not have a complete carnivore guild as lions and cheetahs are absent. The presence of a dominant predator like lions, could greatly affect the activity patterns of the rest of the guild present whereas the presence of cheetahs might have a more subtle and indirect effect (*Vanak et al., 2013*). Furthermore, the inclusion of data from brown hyenas at artificial waterholes would contribute to the full understanding of temporal partitioning in a carnivore guild at artificial waterholes. Even without the complete carnivore guild, artificial waterholes may still be a point of conflict among species, especially for subordinate and already threatened African wild dogs, with only around 500 individuals left in South Africa (*Nicholson et al., 2020*).

The study's focus on artificial waterholes raises questions about whether the patterns observed are unique to these locations or apply to natural water sources as well. The elongated nature of rivers and springs, which occupy a greater spatial extent, may provide a larger area for the avoidance of interspecific conflict, and eliminate the need for temporal segregation. Further research is necessary to determine whether the lack of temporal partitioning within the carnivore guild is unique to artificial waterholes. If this is the case, it could have major implications on future water management in nature reserves and national parks.

In our study we found high degree of temporal overlap in a large carnivore guild at artificial waterholes with no segregation in activity patterns between members of the carnivore guild at artificial water holes. Temporal segregations on roads and game trails were only found in species pairings with African wild dogs.

With the creation of artificial waterholes, the dynamics of animal space-use changed (*Purdon & Van Aarde, 2017*). In some instances, it had negative impact on the surrounding vegetation (*Brits, Van Rooyen & Van Rooyen, 2002*), but it has also brought with it predictability of when and where animals are likely to be encountered (*Sutherland,*

*Ndlovu & Pérez-Rodríguez, 2018*). Artificial waterholes are favoured spots for tourists and hunters, but also poachers who poison the water. Poisoning of waterholes for a target species, like African elephants (*Loxodonta africana*) by ivory poachers, can have cascading effects on all wildlife that use the waterhole (*Mzumara, Perrin & Downs, 2016*; *Ogada, Botha & Shaw, 2016*). The decline of African vulture species is well documented to be connected to active poisoning (*Ogada, Botha & Shaw, 2016*), but large carnivores who frequent artificial waterholes are also at risk although this is poorly documented in Africa (*Olea et al., 2022*). African wild dogs being one of the most endangered carnivores in Africa is of particular concern in this regard, because although they can travel great distances quickly and have been found not to be particularly associated with water source in Hwange National Park (*Ndaimani et al., 2016*), they do visit the artificial water holes relatively frequently as seen in this study. The questions remaining are how prone large carnivores are to poisoning at artificial waterholes and how much do they overlap with other large carnivores at natural water sources.

## ACKNOWLEDGEMENTS

We thank Rieker Botha who provided guidance and much needed knowledge on Maremani Nature Reserve. We would like to pay our gratitude and respects to Dr. Solomon Joubert (1941–2022), a giant pillar in conservation and who provided topical and intellectual discussions for the generation of ideas for this article. We would also like to thank the two anonymous reviewers for providing their helpful comments on how we could improve our manuscript.

### Funding

This work was supported by Aage V. Jensen Charity Foundation who permitted us to collect data on Maremani Nature Reserve and funded research vehicle, dent repairs, fuel, and accommodation for parts of the survey. Lourens Swanepoel received funding from the South African National Research foundation (NRF) for the Snapshot safari project and the DNRF-SARChI Chair in Biodiversity Value and Change, University of Venda. Linnea Worsøe Havmøller received funding from the European Union's Horizon 2020 research and innovation programme under the Marie Skłodowska-Curie grant agreement No. 801199. Rasmus Worsøe Havmøller was supported by research grant 36069 from VILLUM FONDEN. The funders had no role in study design, data collection and analysis, decision to publish, or preparation of the manuscript.

### Grant Disclosures

The following grant information was disclosed by the authors:
Maremani Nature Reserve.
South African National Research foundation (NRF).
DNRF-SARChI Chair in Biodiversity Value and Change, University of Venda.

European Union's Horizon 2020 Research and Innovation Programme: 801199.
VILLUM FONDEN: 36069.

## Competing Interests

The authors declare that they have no competing interests. Gigi Van Zyl is as a volunteer at Maremani Game Reserve and does not receive salary or payment of any form. Gigi has been in charge of volunteer camera trap monitoring of large carnivores at waterholes in Maremani Game Reserve.

## Author Contributions

- Charlotte Krag conceived and designed the experiments, performed the experiments, analyzed the data, prepared figures and/or tables, authored or reviewed drafts of the article, and approved the final draft.
- Linnea Worsøe Havmøller conceived and designed the experiments, analyzed the data, prepared figures and/or tables, authored or reviewed drafts of the article, and approved the final draft.
- Lourens Swanepoel conceived and designed the experiments, performed the experiments, analyzed the data, prepared figures and/or tables, authored or reviewed drafts of the article, and approved the final draft.
- Gigi Van Zyl conceived and designed the experiments, performed the experiments, authored or reviewed drafts of the article, and approved the final draft.
- Peter Rask Møller conceived and designed the experiments, prepared figures and/or tables, authored or reviewed drafts of the article, and approved the final draft.
- Rasmus Worsøe Havmøller conceived and designed the experiments, analyzed the data, prepared figures and/or tables, authored or reviewed drafts of the article, and approved the final draft.

## Animal Ethics

The following information was supplied relating to ethical approvals (*i.e.*, approving body and any reference numbers):

Maremani Game Reserve is a private reserve with no public access owned by the Maremani Nature Reserve Ltd. Camera trapping was only performed on private property with informal permission from the landowner, formal letter now uploaded. All photos of humans have been deleted and is not used in this analyses.

## Data Availability

The raw data and R script are available in the Supplemental Files.

## Supplemental Information

Supplemental information for this article can be found online at http://dx.doi.org/10.7717/peerj.15253#supplemental-information.

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
