# Peer review of "Impact of artificial waterholes on temporal partitioning in a carnivore guild: a comparison of activity patterns at artificial waterholes to roads and trails"

_PeerJ, doi:10.7717/peerj.15253_

## Round 0.1 · original submission · Major Revisions

I am pleased to write the authors to inform them that after reviewing this manuscript both reviewers, and myself, feel this study has some solid merit and could certainly be considered for publication. However, like all articles, there is some room for improvement – particularly with respect to clarity in the manuscript’s writing, clearing up some ambiguity in methodological decision and descriptions, and potentially revisiting the analysis to improve what conclusions can be drawn.

Overall, the comments and suggestions are not overall arduous, and should the authors be able to address them, to further increase the readability and interpretation of information, then I do feel this could certain be a publishable article.

Reveiwer 1’s comment mostly about increasing clarity with respect to the information you provide and the rationales behind some of the methodological decisions made. These includes adding information about the difference in activity periods between Africa Wild Dogs and the other predators within the Introduction (to set up the idea later discuss in the paper). They also note that it would be useful to provide the reader some information on the various density of these predator populations. They also cover a number of other aspects that could be useful to convey to the readers a bit more clearly (see review). One thing that both Reviewer 1 and I noted as seeming a bit odd, was the statements made on lines 199-202 “Because of the high numbers of brown hyenas in recent years at our study site, it was not seen as a conservation priority to annotate data of brown hyenas at artificial waterholes. Therefore, we cannot know if there would exist temporal partitioning between brown hyenas and the rest of the carnivore guild at artificial waterholes.”. Across the rest of the paper, including the immediate sentence before this, the value of examining brown hyenas is stated (which I agree with). So why not annotate and analysis that data as well. This is especially true if they have increased their numbers in recent years. The fact they are not seen as a conservation priority is irrelevant to the ecological interactions occurring, in which this species is taking part in. As this is a paper examining spatiotemporal overlap, partitioning, and segregation, and brown hyenas will impact this. If the data exists and can be found and included, it should be.

Reviewer 2’s comment also has to do with providing some more context to methodological choices and suggests examining an alternative analytical approach (time-to-encounter analysis) that could yield some more detailed information and provide a stronger test of spatiotemporal overlap (see review). Should you consider this suggestion and feel that your current approach provides a more defensible and thorough analysis of the data, and thus choose not to conduct a time-to-encounter analysis, then I would like to know your rationale. Additionally, they note that it would be useful to give some better context regarding “artificial waterholes”. I know locally they are a common and well known landscape feature and term. But as this is going out to a global audience, I can see the value in defining and contextualizing them, as Reveiwer 2 suggests.

Two methodological choices that caught both Reviewer 2 and my eye, related to the survey timing and study design. Firstly the survey duration varies widely: Survey 1 = ~210 days, Surveys 2 = ~90 days, and Survey 3 = ~ 3650 days. Reviewer 2 notes that over such a long, and varied, timeframe this could impact the data collected. I think this is a fair point and should be discussed or at least contextualized for the reader. That said, I do understand that this data was collated into the two categories of near or away from the artificial watering hole, and that as long as the survey effort between these categories is equivalent then it should be okay. But I cannot seem to find a statement saying it is. The other methodological choice that seems like it could be sticky, if not properly explained, is the choice to set the “away from artificial watering hole” cameras only 270 m away. This does seem like it is still pretty nearby. Albeit not at or directly beside. However, for all of these species that distance is pretty low. What was the reason for this choice from a spatial ecology perspective? Was it based on previous research? Was it set to some mean daily movement distance averaged across the taxa being examine? How did you come to land on 270 m from an experimental design and validity point of view?

Additionally, although the writing quality is relatively high, I would still encourage the authors to pay attention to tighten up a little more of the writing structure. Look to: (1) strengthen some of the topic sentences for serval paragraph (e.g., notably the first sentence of the Introduction that sets the entire tone and context of the paper), (2) work in some stronger segues between paragraphs within your concluding sentences, (3) try to minimize qualifiers at the beginning of sentences where you can (e.g., “However,”, “For example,”), and (4) keep a keen eye out for typos. Once the reviewer suggestions are incorporated and the content is sound, I will do another pass for these sorts of things, but better to get a drop on it now during revisions to make that step easier down the line.

Overall, I hope you find all of these comments and suggestions helpful. Our goal is to provide feedback so that you are able to continue to improve and strengthen the write up of your work. In doing so, we are endeavoring to make sure your research is sound, solid, defensible, and able to be enjoyed, understood, and cited by a wide range of readers. I am looking forward to seeing your revised manuscript. Keep well!

Reviewer 1 ·

Basic reporting

Use of the English language throughout the manuscript was of a high quality. The paper is structured in a logical order and references were properly used throughout. Hypotheses were clear although not very exciting.

Experimental design

Research questions well framed, and it does try to fill a knowledge gap. It would have been nice if the authors would have included natural waterholes in their survey as well, could have been a nice comparison. In addition, it is general knowledge that wild dog have a different activity pattern compared to the other large carnivores, would have been useful to have stressed that somewhere in the introduction / methods. In addition, some more information on densities in the area, if possible , would be a great addition to the manuscript.

Validity of the findings

The results and discussion are logically linked. Benefit to literature Is clearly stated and conclusions are also logical.

Additional comments

Abstract
- P1, 14. Temporal partitioning is not the only factor enabling co-existence between large carnivores. What about diet? What about spatial partitioning? You should at least mention these as other factors that might explain co-existence. I understand that dynamics at a waterhole are different, since resource is fixed, but this sentence should be rephrased.

Introduction
- Line 60, become -> becomes
- Line 65, but see Edwards et al etc. Why the “ but see” . Do Edwards et al and Atwood et al contradict this, or support this statement? If support, remove the “but see”.
Line 78-81. Spotted hyena are known to be dominant over wild dog, however encounters between leopard + brown hyena and wild dog are not one sided and frequently wild dog are dominant over both carnivores. Stress that Hayward & Slotow only looked at the dominance of spotted hyena over wild dog, not the other carnivores.
- All in all, it is worth mentioning that wild dogs are mostly diurnal and not nocturnal and that because of this you expect already temporal partitioning. However, there is other research that suggests that wild dog might be more nocturnal than previously assumed.


Methods
- Did a camera trap station consists out of 1 or 2 camera traps, please specify?
- Is there a resident wild dog pack in the study area? Or transient?

Results
- Figure 2, the figure is very full. It might be better to move the overlap coefficient and the range to the caption for every plot. Same for figure 3 and 4.

Discussion
- Research by Vissia et al (2022) supports your findings that there is no significant difference in activity patterns between leopard and spotted hyena and brown hyena

Vissia, S., Fattebert, J., & van Langevelde, F. (2022). Leopard density and interspecific spatiotemporal interactions in a hyena‐dominated landscape. Ecology and Evolution, 12(10), e9365. https://doi.org/10.1002/ece3.9365

- Line 199-200. Because of the good numbers of brown hyena in the study area, it was decided that they wouldn’t be included in the analysis?
- Line 216-217, how would the presence of a subordinate carnivore affect activity patterns of other guild members? For a dominant carnivore like lion this is obvious, but for a subordinate carnivore like cheetah?
- Line 220, was there another reason, besides logistics I guess, why natural water sources were not included in the survey to make a comparison between artificial and natural water sources?
- Line 222-223 and 225-228, why would there be a difference between natural and artificial water sources? Please elaborate.
-

·

Basic reporting

This study addressed the temporal niche partitioning between competitive large carnivores in African (semi-)arid ecosystems, by camera-trapping survey. This study used very simple but validate methods for assessment of interspecific temporal separations that have been used in many previous studies recently (but also there's a room for improvement in the analytical methods; for more details, see 2. Expeimental design). The authors found potential temporal segregations between subordinate (i.e., African painted dogs) and sperior species for avoiding competitions at waterholes, indicating their temporal partitioning would be important for their coexistences. This is a good case study demonstrated that niche partitioning is crucial role for sympatry of cometitive species under restricted resrouce conditions. Therefore, this study is potential to publish in the jouranl.
I have some concerns in study sites and experimental designs, for understandability of readers (please see the next review section). If the authors improve them and/or add more details, it is acceptable to publish.

Experimental design

For the present text, I have some concerns as following:

1. I'm sorry but I don't have good knowledge for "artificial waterhole" in Africa. However, this may be common for many readers who are not familiar to this region. Specifically, I have a concern whether/how was the anthropogenic impacts at the "artificial" waterholes and how different those impacts between at the waterhole sites and sites far from waterholes. Artifical waterholes is onaly for use by wild animals or any use by local people (e.g., agriculture or livestock drinking, or hunting)? Generally, human activities affect diel activities of wild animals and increase their nocturnality (e.g., Gaynor et al. 2018 Science 360, doi: 10.1126/science.aar7121) and altering interspecific niche paritioning (e.g., Seveque et al. 2020 Biol. Rev. 95, doi: 10.1111/brv.12635). Please clarify how were (NOT) the human acitivites (types and intesities) in your study area and whether there were (NOT) the potential impacts on the animal behaviors.

2. The authors used the data from three different sampling periods (L103-111). Specifically, "Survey three" was conducted for ca. 10 years, while the remaining two sessions (Survey one and two) was shorter peridos than it. Why did authors do that different sampling sessions? There might be large environmental changes (e.g., drought or much yearly rain) during the former (i.e., 10 years sampling in Survey three session) and it could be affect the intesity/frequency of utilizations of waterholes by animals? The authors should be clarify that there were little problems/effects for using the different sampled data for direct comaprisons.

3. This is an optional comment: currently more detailed analysis were recommended to assess interspecific temporal niche partitioning. For example, Karanth et al. (2017) in Proc. R. Soc. B 284 (doi: 10.1098/rspb.2016.1860) suggested use of time-to-encouter analysis. We also currently assessed and demonstrated that different analytical methods showed different results of niche overlap/separtions (Watabe et al. 2022 in Sci. Rep. 12, doi: 10.1038/s41598-022-16020-w). Other methods, like time-to-encouter analysis, would be more appropriate and powerful tool to clarify temporal separations/overlaps in your focal carnivore guilds.
NOTE: this is optional and just suggestion. If the authors decide (not) to add new analysis in revisions, I respect their decision.

Validity of the findings

I have only one comment in this section as below.

The distance between sites at wateholes and sites far from waterholes were > 270m (L113). It looks not so long distance for mobile species, e.g., African painted dogs, and they may easily move between the two compared sites. If so, they can separate not only temporally but also spatially: e.g., when the subordinate species recognized presece of dominant competitor at a waterhole site, they can escape from there and can visit at another site. To support your findings/conclusions (importance of temporal segregations), applying spatiotemporal analysis, e.g., time-to-encouter analysis (see above section), would be more appropriate. Or, at least, you need to show that they did not separate at site-levels (i.e., co-occur equally at any observed sites). For example, applying Sorrenson's or Pianaka's index is easy way to assess it.

Additional comments

I found a minor point for grammatical revision:

L196 "also seen in seen here" > Mybe "also seen here"?

---

## Round 0.2 · Minor Revisions

I am quite pleased to see the authors have done a good job addressing the reviewers' previous comments, and both the reviewers and I think the quality of the research is there and should merit acceptance. There is, however, a number of typos, some structural issues, and a bungled in-text citation issue, that will need to be addressed. As well as a few great suggestions from the reviewers. If these can be fixed, and a thorough polishing of the text can be done, then I think we will be off to the races.

Please go over all of the reviewers' comments and suggestions during your revision. I have also gone through and tried to catch as many editorial fixes, as I could see, and I will email you a copy of a tracked changes file for ease of editing. I am certain the reviewers and I have not caught it all, so do please pay close attention to recurring issues like in-text citations with author initials and so on. Some of this may just be a problem with a citation manager software, but just work through it manually to be certain.

Reviewer 1 ·

Basic reporting

Dear authors,

Thank you for re-submitting your manuscript and the opportunity to review your manuscript again. It is clear the authors have taken our feedback into consideration and made appropriate changes. The manuscript is more scientifically sound and better structured and decisions that were made are better explained.

I do have some additional comments which could help with improving the manuscript.

Overall. The English language is properly used throughout the manuscript, however I suggest that the authors do one more spelling / grammar check to make sure small mistakes are removed from the manuscript.
e.g. L105, the source annonymous is spelled with two N, while one is needed.
L147, cameras were non-paired instead of non-pared

Thanks for the feedback on figure 2-3-4 and your arguments to not change the lay-out.

Experimental design

I am still unsure about the camera trap that was used at 270 m away from the artificial watering hole and is classified as such. You try to remedy this by using leopard data from Havmøller et al. (2019) and by arguing this is 25 % of the value found for mean leopard daily movements. However, the study area in the referred to study was in the Udzungwa mountains in Tanzania, a completely different landscape than the landscape in this study. So, how realistic is it to expect similar daily movement numbers.

Validity of the findings

Lastly, I appreciate that the authors clearly have tried to restructure the discussion to include all information that might be important regarding this study. However I would advise the authors to compress the discussion as the length of this section is now out of proportion compared to the other sections.

Additional comments

Another issue I have is the confusion about brown hyena numbers in the manuscript. I understand that it was completely out of your control that the volunteer deleted all the data regarding brown hyena. However, you stated in the first manuscript lines 199-200 that brown hyena numbers were high in recent years. In the new manuscript , lines 112-113, you now state that brown hyenas have experienced a decline in numbers since 2016. Which one is correct? Interestingly, Is it a coincidence that it corresponds with the re-introduction of spotted hyena in the area?

·

Basic reporting

The revised manuscript is appropriately improved, responding to comments or suggestions from reviewers. However, I have found some minor points to need additional corrections or revisions. If they were improved by the authors, I think it is acceptable for publication.

Minor points:
L55: Remove some abbreviations of first names of cited literature (i.e., M. W. and R.).

L201&202: Move "e.g." inside the following parethesis, i.e., like "(e.g., Atwood et al. 2011..."

L207-209: I cannot understand why the authors say this sentence here? I recommend them to remove it or move to appropriate part.

L 231&235: Again, remove "S."

L252: Again, remove "M. W." and "R."

L258: "tape" > "type" is maybe right?

L258&262: Again, remove "R."

L287: Use plural like "both spotted and brown hyenas".

L302: Again remove, R. W.

L304: Again, remove "M. W." and "R."

L348: Change from "?" to "."

L404-406: "Edwars et al. 2015" is duplicated. Remove the later.

L410-416: I think you need to use Havmoller et al. (2020a) & (2020b) in this text. They are different references from same author.

L429-434: Again, the referecen is duplicated.

L442-445: Again, the referecen is duplicated.

Experimental design

no comment

Validity of the findings

no comment

---

## Round 0.3 · accepted · Accept

I think the authors have done a great job revising this study and it associated write up. From a science standpoint and a paper setup perspective I am happy to accept this, and want to congratulation the authors on a job well done! I hope people find this study useful and interesting.

There are still a few typos, so please do read carefully during your proofing stage. For example: (1) please place all "e.g.," and "i.e." in parentheses, and it's use on line 16 does not appear to make sense, (2) you are missing an oxford comma in a number of places, like after "leopards" on line 95 (also in lists on lines 19, 47, 113, 198, and 213) and the word "accommodation" is missing an "m" on line 334. There is likely a few more. So just do keep an eye out. From an academic editor standpoint, I am happy to recommend this is accepted. But please do bring out your "fine toothed comb" during proofing.